# Multi-method proof-of-concept evaluation for *R2Play*: a novel multi-domain return-to-play assessment tool for concussion

Josh Shore[1,2], Pavreet Gill[1], Danielle DuPlessis[1,3], Andrew Lovell[1], Andrea Hickling[1,4], Emily Lam[1], Fanny Hotzé[1], Elaine Biddiss[1,2,5], Shannon E. Scratch[1,2,6]*

1 Bloorview Research Institute, Holland Bloorview Kids Rehabilitation Hospital, Toronto, Ontario, Canada, 2 Rehabilitation Sciences Institute, Temerty Faculty of Medicine, University of Toronto, Toronto, Ontario, Canada, 3 Department of Psychology, York University, Toronto, Ontario, Canada, 4 Department of Occupational Sciences & Occupational Therapy, University of Toronto, Toronto, Ontario, Canada, 5 Institute of Biomedical Engineering, University of Toronto, Toronto, Ontario, Canada, 6 Department of Pediatrics, Temerty Faculty of Medicine, University of Toronto, Toronto, Ontario, Canada

* sscratch@hollandbloorview.ca

## Abstract

Return-to-play (RtoP) clearance after concussion typically involves single- and dual-task assessments that do not reflect the speed or complexity of sport. We developed *R2Play*, a dynamic multi-domain assessment tool for concussion. This study aimed to (1) demonstrate proof of concept for *R2Play* by evaluating alignment with design objectives (easy to use, fun, sport-like, clinically valuable, resource efficient, and flexible); and (2) document subsequent iterations to *R2Play* design. A multi-method evaluation was performed wherein clinicians were paired with youth to test *R2Play* together and complete separate semi-structured interviews. Quantitative metrics included the System Usability Scale (SUS), heart rate (HR), ratings of perceived exertion (RPE), assessment durations, and *R2Play* completion times, errors, and multi-task cost scores (changes in performance with the introduction of new challenges). Interviews explored perspectives on design objectives, analyzed using content analysis. Participants included five clinicians (n = 2 occupational therapy; n = 1 physiotherapy; n = 1 athletic therapy; n = 1 medicine) and 10 youth (ages 10–22 years). Assessments took 30–40 minutes despite minor technical challenges (e.g., unresponsive equipment). Clinician-rated usability was good-to-excellent (SUS = 81 ± 8.4; 95% CI: 73.6, 88.4) and youth reported that instructions were easy to follow. Moderate-to-high-intensity exertion was achieved (peak HR = 80 ± 11% age-predicted maximal; 95% CI: 77.4%, 88.5%). Multi-task cost scores reflected some aspects of hypothesized level demand loading. Clinicians described *R2Play* as potentially valuable to assess sport tolerance and enable rich observations of multi-domain skill integration. Tables were constructed to map study findings onto design iterations. This study supports proof-of-concept for *R2Play*, a new

**Data availability statement:** Given the aims and scope of this preliminary proof of concept study, most relevant data can be found within the manuscript and Supporting Information files. Additional data that support the findings of this study cannot be shared publicly due to research ethics policies because research participants did not provide consent for public sharing of their data. In compliance with institutional and ethical standards of operation, additional data may be made available on request to the Research Ethics Office, Holland Bloorview Kids Rehabilitation Hospital, 150 Kilgour Road, Toronto, ON M4G 1R8. Tel: (416) 425-6220, ext. 3161. E-mail: researchethics-board@hollandbloorview.ca.

**Funding:** This work was supported by a project grant from the Centre for Leadership at Holland Bloorview Kids Rehabilitation Hospital (https://hollandbloorview.ca/) awarded to authors EB and SS, Holland Family Professorship in Acquired Brain Injury (SS), and Bloorview Children's Hospital Foundation Chair in Pediatric Rehabilitation (EB). JS received graduate student funding from the Temerty Faculty of Medicine at the University of Toronto (https://rhse.temertymedicine.utoronto.ca/), Ontario Graduate Scholarship (https://rhse.temerty-medicine.utoronto.ca/ontario-graduate-schol-arships), and Canadian Institutes of Health Research (https://cihr-irsc.gc.ca/e/38887.html). DD was supported by a Holland Bloorview Graduate Foundation award (https://hollandbloorview.ca/) and an Ontario Graduate Scholarship through the Rehabilitation Sciences Institute at the University of Toronto (https://rhse.temertymedicine.utoronto.ca/ontario-graduate-scholarships). The funders had no role in study design, data collection and analysis, decision to publish, or preparation of the manuscript.

**Competing interests:** The authors have declared that no competing interests exist.

multi-domain concussion assessment tool, and identified areas for improvement, which has informed changes to the design of *R2Play* before broader evaluation among youth post-concussion.

---

## Author summary

Traditional concussion assessments do not reflect the speed or complexity of sports. Our team developed *R2Play*, a dynamic, multi-domain, digital concussion assessment tool designed to help simulate sport demands within clinics by combining physical, cognitive, and sensory skills. In this study, we evaluated alignment of the *R2Play* prototype with initial design objectives that it should be easy to use, fun, sport-like, informative, resource efficient, and flexible. We invited five clinicians and 10 youth (ages 10–22 years) to test *R2Play* and provide feedback through qualitative interviews. We also collected quantitative data including usability scores, heart rate, and *R2Play* performance metrics (completion times and accuracy). We found that *R2Play* was easy to use and involved moderate-to-high intensity exertion, with participants describing the assessment as engaging, enjoyable, and reflective of sport. Clinicians recognized the value of *R2Play* for observing complex skill integration to monitor recovery, suggesting a potential role for novel multi-domain assessments in practice. This study supports proof-of-concept for *R2Play* as a promising new multidomain assessment tool for concussion and informed its refinement through design iterations. It also shows how creative application of low-cost digital technology can stimulate clinical innovation and demonstrates the value of early small-scale testing with end-users of digital health tools.

## Introduction

Concussion is a common sport injury [1,2]. Due to ongoing neurological and psychosocial development, concussion manifests uniquely in youth, requiring age-appropriate approaches to assessment and rehabilitation [3]. Proper management of concussion and the return-to-play (RtoP) process is essential, as misdiagnosis or premature RtoP may have significant consequences including prolonged recovery or recurrent concussions [4,5], subsequent orthopedic injury [6,7], and in rare cases catastrophic brain injury resulting from secondary impact [8,9]. Best practice guidelines recommend a stepwise approach to RtoP involving progression through stages of increasing exercise complexity [3]. Importantly, thorough clinical assessment is required before resuming high-risk activities to ensure symptom resolution and restoration of physical and cognitive function at rest and with exertional challenges [3]. Legislation has thus been introduced in several jurisdictions to mandate RtoP clearance from a healthcare provider among youth after concussion [10,11]. However, evaluating concussion recovery and readiness to RtoP remains challenging, as there

are limited evidence-based standards for decision making and substantial variability in criteria utilized within the research and clinical communities [4,12–15].

Emergent research has demonstrated a need for more complex assessments that simultaneously target multiple clinical domains in order to uncover subtle lingering changes post-concussion [16–18]. While standard single-domain physical and cognitive measures often normalize within a few weeks post-injury [19], persistent sub-clinical deficits in performance are apparent under concurrent cognitive-motor dual-task and perceptual-motor integration paradigms [16–20]. Such differences are more pronounced as physical and cognitive task complexity increases [21,22]. Thus, it is problematic that current dual-task assessments for concussion, primarily comprised of walking paired with simple cognitive tasks (e.g., spelling, arithmetic), do not fully reflect the speed or complexity of processing in sport, which requires continuous integration of physical, cognitive, and perceptual domains within a rapidly evolving environment [23–25]. Additionally, clinical implementation of existing dual-task assessments may also be limited by practical challenges related to test administration and scoring both physical and cognitive performance accurately within clinical environments [23]. Digital technologies such as smartphones and wearable devices may help translate existing dual-task assessments to clinic but these approaches remain limited in ecological relevance to sport [26,27]. An opportunity thus exists for user-driven technologies to help achieve the sport-like complexity and assessment sophistication required to detect subtle neurophysiological and neurocognitive deficits while ensuring feasibility of implementation in clinical settings.

Our interdisciplinary team sought to address these gaps with *R2Play*, a dynamic multi-domain assessment tool designed to support RtoP decision making among youth with concussion by simulating the demands of sport within a clinical setting. *R2Play* was developed through a rigorous user-centered design approach involving a scoping review of existing multi-domain concussion assessments [23], discussion within our clinically-integrated research team, qualitative needs-assessment interviews with practicing clinicians and sport coaches, and iterative usability testing with clinicians [28]. Following this process, the *R2Play* concept (Fig 1) was established as a series of low-cost touchscreen tablets that youth run between to connect in alphanumeric order (1-A-2-B-3-C), similar to the Trail Making Test [29], advancing through several levels with layered physical, cognitive, and perceptual challenges. A separate laptop operates the *R2Play* clinician interface software, which is used to control the tablets, administer the assessment, monitor clinical measures (heart rate, perceived exertion, concussion symptoms), record clinical notes, and track performance via level completion

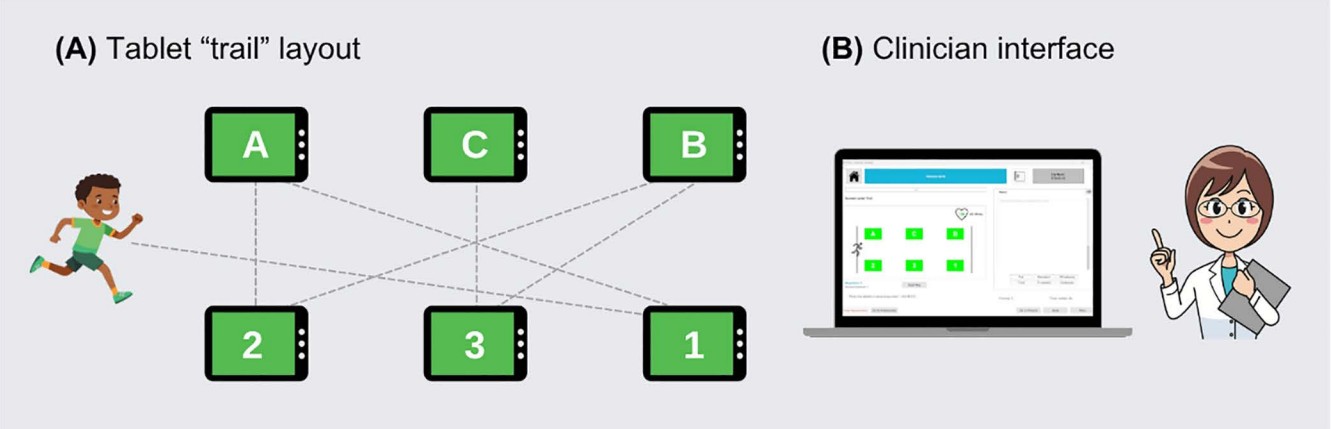

**Fig 1. *R2Play* assessment concept. (A)** Tablet "trail" layout that youth run between to connect in alphanumerical sequence (1-A-2-B-3-C), and **(B)** Laptop operating clinician interface used to control tablets, administer *R2Play* assessment, monitor clinical measures, and track performance. Figure prepared using open-source assets from openclipart.org.

times, errors, and multi-domain cost scores reflecting changes in performance with the addition of new challenges. A complete description of the *R2Play* development process is available in DuPlessis et al. [28].

As a first step towards evaluating *R2Play*, and in keeping with our user-centered design approach, we invited clinicians and youth sport participants to test *R2Play* and provide feedback to inform its iterative refinement. The primary aim of this study was to demonstrate proof of concept for *R2Play* by evaluating alignment of the prototype with initial design objectives established during the development phase, including that *R2Play* should be easy to use, fun, sport-like, clinically informative, resource efficient, and flexible. A secondary aim was to document subsequent iterations to the design of *R2Play* based on this initial testing.

## Materials and methods

### Design

A multi-method approach was undertaken to evaluate alignment of the *R2Play* prototype with initial design objectives. Clinicians and youth were paired to test the prototype together in person and provide feedback through semi-structured interviews. Paired testing was used to create more realistic conditions for participants to reflect on future real-world use of *R2Play* in clinical practice. Data captured by the *R2Play* system included performance measures (completion times, errors) and clinical parameters (symptoms, heart rate, perceived exertion). Table 1 outlines the design objectives and preliminary evaluation methods for this proof-of-concept study. Design objectives were established during the original development of *R2Play* and were informed by a scoping review of existing multi-domain concussion assessments, [23] literature on digital health technology adoption, and experiences of the interdisciplinary design team, including discussion with clinical, engineering, and family partners.[28].The specific rationale for each design objective is described in a previous publication [26]. Two objectives were modified to align with the scope of this preliminary proof-of-concept study, as described in the footnote of Table 1.

**Table 1. Proof-of-concept evaluation of *R2Play* design objectives from DuPlessis et al., 2022 [28].**

| Objective | Description | Evaluation methods |
|---|---|---|
| Ease of use | Assessment should be easy for youth athletes to understand, and the system should minimize the cognitive load on clinicians during administration and scoring. | • System Usability Scale<br>• Qualitative interviews (clinician and youth) |
| Fun for youth athletes | Task should be engaging and fun for youth athletes ages 10–25 years old. | • Retention of participants through entire assessment protocol<br>• Qualitative interviews (clinicians and youth) |
| Sport-like | Assessment should require integration across physical, cognitive, perceptual, and socio-emotional domains while responding to a dynamic environment. | • Physiological and perceived exertion attained by each participant (mean and peak HR, RPE)<br>• Qualitative interviews (clinicians and youth) |
| Potential clinical value* | Clinicians should see value in the assessment to provide useful information for informing RtoP decision making in practice | • Changes in reported concussion-related symptoms requiring assessment cessation or modification<br>• Degree to which raw *R2Play* results and cost scores reflect hypothesized level loading (Table 2)<br>• Qualitative interviews (clinicians) |
| Resource efficiency† | Resource requirements (e.g., time, cost) should be minimized as much as possible while maintaining functionality. | • Cost of system prototype<br>• Duration of assessments<br>• Qualitative interviews (clinicians) |
| Flexibility | System should be usable across different clinical spaces and customizable to suit the needs and abilities of individual youth. | • Participant characteristics<br>• Protocol deviations and required adaptations<br>• Qualitative interviews (clinicians) |

RtoP, return-to-play. HR, heart rate. RPE, rating of perceived exertion. *The original *clinically informative* design objective was changed to *potential clinical value* to reflect the preliminary nature of the *R2Play* assessment and scoring parameters in its current stage of development. †*Resource efficiency* was broadened from the original *low-cost* objective to encompass other aspects of implementation (e.g., time).

### Ethics statement

Ethics approval was obtained from Holland Bloorview Kids Rehabilitation Hospital (REB #20–099) and the University of Toronto (REB #40694). All clinician participants provided informed written consent prior to commencing study activities. Per institutional policy, youth participants' capacity to consent to research participation was assessed by the research team prior to study enrolment. If the youth could not demonstrate capacity, a parent or guardian would be asked to provide informed written consent on behalf of their child, with the youth providing written assent. In the final study sample, all youth demonstrated capacity and thus provided their own informed written consent prior to commencing study activities.

### Participants

Five clinicians and 10 youth were recruited to test and provide feedback on the prototype. This sample size aligns with technology usability literature and was expected to provide sufficient data to address the aims of this study and guide system refinement [30]. We conducted this proof-of-concept work with youth who did not have recent concussion history to enable rapid testing and ensure suitability and safety before use with a clinical population. Testing *R2Play* among youth with recent experience of concussion and gathering their feedback will be a key future direction.

Clinician participants recruited from our institution and the community, were required to have at least one year of experience working with children or adolescents and a caseload including RtoP after concussion. Youth were eligible if they were 10–25 years old, members of a sport team with ≥3 hours weekly commitment (excluding COVID-19 shutdowns) and had normal or corrected-to-normal vision and hearing. Exclusion criteria included: (1) recent musculoskeletal injury (within previous 10 days), pre-existing physical condition, or neurological disorder which could be aggravated by exercise (including concussion); (2) visual, cognitive, or auditory disabilities that could affect *R2Play* performance; and (3) pre-existing cardiovascular conditions that could be exacerbated by exercise or cause abnormal electrocardiogram. Youth were recruited through institutional advertisements, internal clinical programming, social media, and word of mouth.

### Study protocol

**Fig 2** summarizes the study protocol. Sessions began with informed consent and collection of demographic information for clinician and youth participants. Clinicians were then given brief (~10 minutes) verbal training on the *R2Play* assessment and interface from a research team member while youth donned a heart rate (HR) monitor and completed physical warm-ups (e.g., jogging, side-shuffles, lunge-and-twist). Youth then completed pre-assessment measures (resting HR and concussion symptom scale) and the *R2Play* assessment, administered by the clinician. Immediately after the *R2Play* assessment, a research team member reviewed results with both participants before conducting individual post-assessment interviews. Due to the novelty of *R2Play*, clinicians were asked to attend two sessions with different youth participants to provide sufficient experience with the system and enable more comprehensive feedback. After the second session, clinicians completed a virtual follow-up interview to reflect on their overall experiences and perspectives regarding *R2Play*.

### Study measures

**Demographic information.** Youth completed a brief demographic form to report their age, gender, and sport participation including weekly hours of training and competition. Clinicians reported their age range, gender, discipline, years of experience, practice sector (e.g., public, private), and age range of clients.

***R2Play* assessment.** ***R2Play* assessment protocol:** The *R2Play* assessment in this study (version 1.0) consisted of four core levels that varied physical and cognitive demand complexity, plus a pre- and post- motor task (**Fig 3**) [28]. The baseline *Number-Letter Level* involved participants running to tap the tablets in alphanumeric sequence per the characters displayed on screen (1-A-2-B-3-C). Two cycles of this six-character sequence were performed in each repetition, for

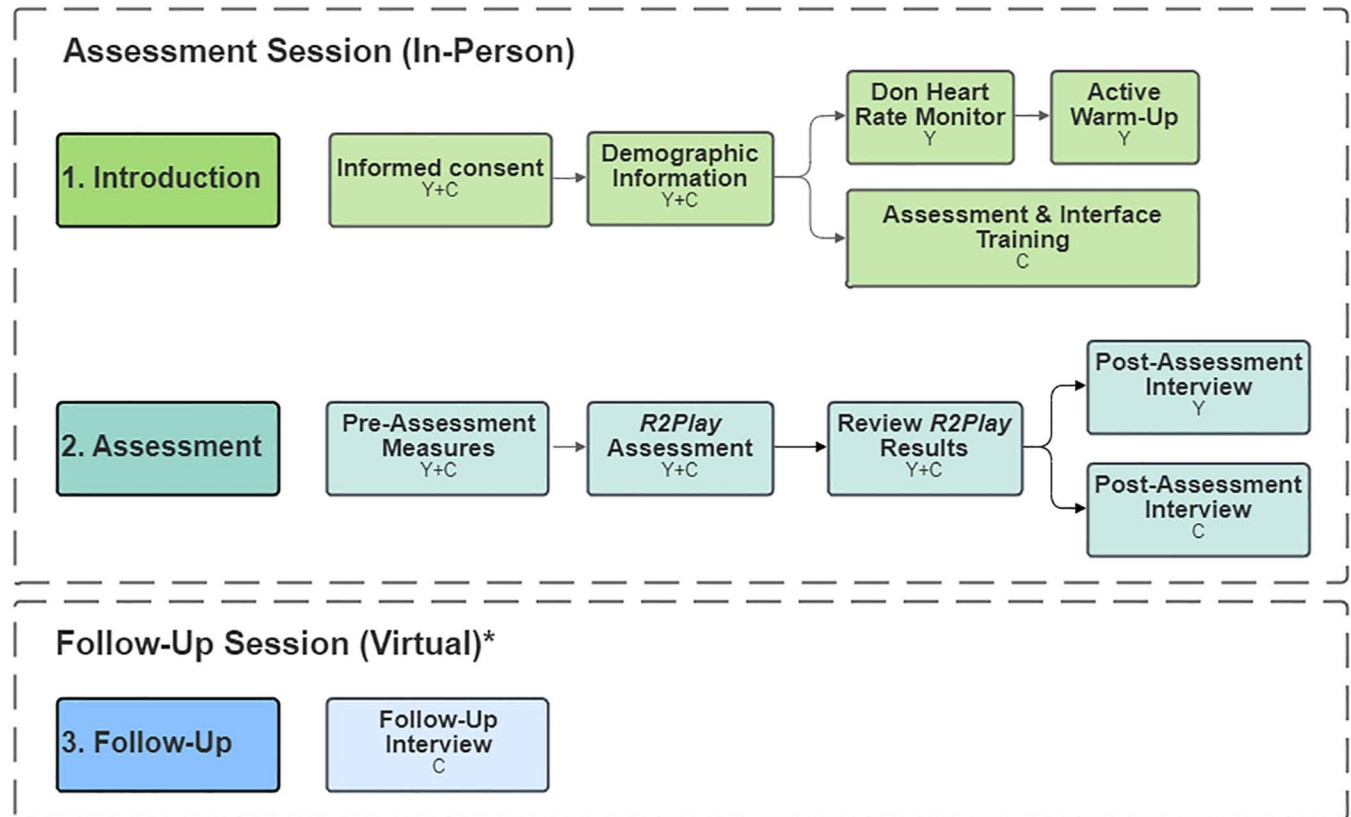

**Fig 2. Study protocol.** C, Clinician participant. Y, Youth participant. *Follow-up session completed within approximately 1 week after clinicians' second assessment session.

a total of 12 tablet selections. In the *Exercise Level*, a beep cue played after the participant tapped one of the tablets in the trail, after which they completed three repetitions of a chosen exercise (jumping jacks, tuck jumps, or burpees) before continuing the trail. The *Go-No-Go Level* introduced response inhibition as participants were instructed to complete the alphanumeric trail using only the letters/numbers displayed with a green background and exclude those with a red background. This rule was flipped in the *Stroop Level*, wherein participants had to select only the red number/letter tablets and exclude the green ones. Trail sequences were programmed such that the added distance between correct tablet taps was standardized to 45m (± 1%) in *Number-Letter* and *Exercise* level repetitions and 30m (± 1%) in *Go-No-Go* and *Stroop* level repetitions (due to red/green tablet omissions). For all levels, distance travelled *per correct tablet selection* was approximately 3.75m.

Each *R2Play* level included four repetitions that introduced layered challenge conditions. The first repetition was the *Standard* condition (no added challenge), while repetitions 2–4 occurred in a randomized order and included another Standard condition, an *Auditory Interference* condition in which background sport noises (e.g., cheers, whistles) played during the trail, and a *Scramble* condition in which the numbers and letters unexpectedly relocated to different tablet positions partway through the trail, engaging perception-action integration loops as participants responded and adapted to the new orientation [18,20].

Levels began with a brief orientation, facilitated by the clinician interface software, in which verbal instructions were provided and participants could watch a video demonstration and practice tracing the number/letter pattern on the computer.

### Motor Trail (Pre)

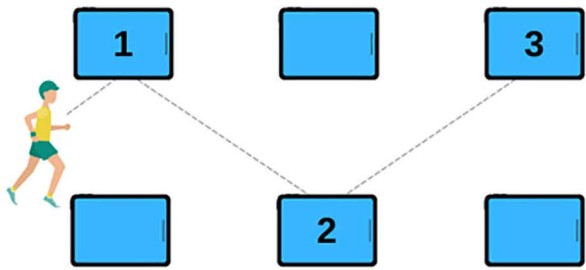

Connect the dots by pressing the tablets in order. Start with the lowest number and finish with the highest number: 1-2-3.

### 1. Number-Letter Level

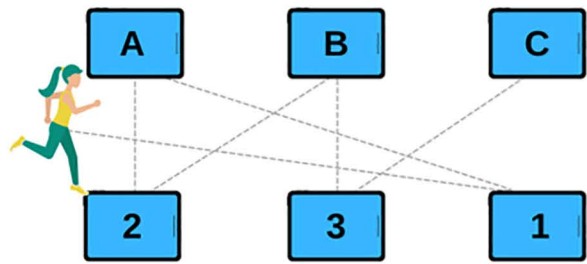

Connect the dots by pressing the tablets in order. Start with the lowest number, then the first letter of the alphabet, and finish with the last letter shown to you: 1-A-2-B-3-C.

### 2. Exercise Level

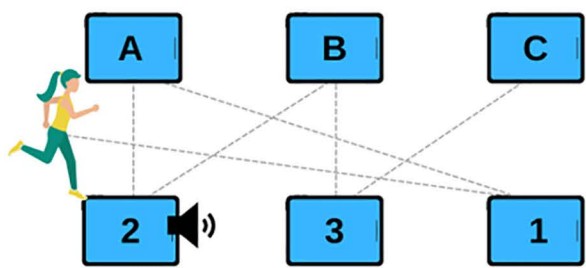

Connect the dots by pressing the tablets in order. This time, when you hear a **beep** noise, do [3 reps of burpees]*.

*Number of reps and exercise type is customizable.

### 3. Go-No-Go Level

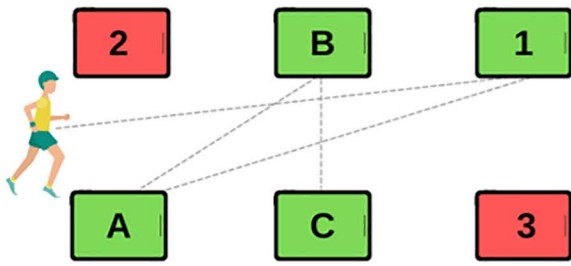

Connect the dots by pressing the tablets in order. This time, some of the tablet screens will be coloured green and some will be coloured red. Connect the dots while pressing only the green tablets and avoiding the red tablets.

### 4. Stroop Level

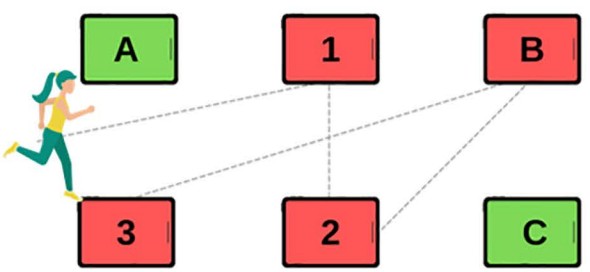

Connect the dots by pressing the tablets in order. This time, some of the tablet screens will be coloured green and some will be coloured red. Connect the dots while pressing only the red tablets and avoiding the green tablets.

### Motor Trail (Post)

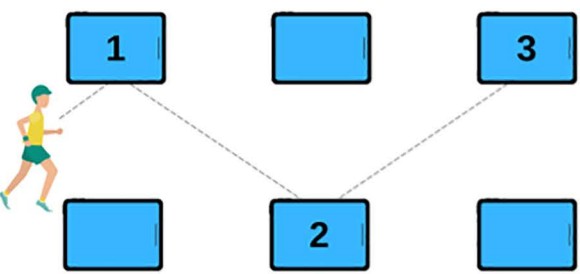

Connect the dots by pressing the tablets in order. Start with the lowest number and finish with the highest number: 1-2-3.

**Fig 3. *R2Play* assessment levels.** The assessment included four core levels that varied physical and cognitive demand complexity, plus a pre- and post- motor task. Image reproduced from DuPlessis et al. [28] published in *Frontiers in Rehabilitation Science* (2022) under CC BY 4.0 license (https://creativecommons.org/licenses/by/4.0/).

After each level, a brief check-in was conducted during which participants reported their rating of perceived exertion and any changes in concussion-related symptoms (*see measures below*) through a dedicated screen in the clinician interface. HR was captured throughout the assessment and streamed to the clinician interface. A video overview of the *R2Play* assessment can be found in S1 Appendix.

***R2Play assessment scoring:*** *R2Play* performance is measured through completion times, errors (i.e., incorrect tablet selections), and HR. Since the Go-No-Go and Stroop levels require fewer tablet selections by omitting those with red/green backgrounds, completion time is calculated relative to the number of correct tablet selections in the level, expressed via the standardized metric of seconds per tablet (seconds/tablet). Using these metrics, multi-domain cost scores (expressed as percentages) are produced that reflect changes in performance between levels and conditions with the introduction of new challenges (**Table 2**). Similar to traditional dual-task cost equations, multi-domain cost scores are calculated by determining the percentage change in completion time, errors, or HR between a baseline level or condition and a more challenging level or condition. Since they each reflect the introduction of a new challenge, positive values are expected for all cost scores (i.e., an increase in time, errors, or HR).

**Exertion measures.** *Physiological exertion:* Youth were outfitted with a PolarH10 chest strap monitor (Polar Electro, Kempele, Finland) to capture their resting HR before the *R2Play* assessment and their active HR throughout the

**Table 2.** *R2Play* multi-domain cost score calculations.

| Cost score | Increasing load | Comparison | Calculation** | Example*** |
|---|---|---|---|---|
| Exertion cost | Physical | Number-Letter level (standard condition) Vs. Exercise level (standard condition)* | [Completion time (Exercise level) – Completion time (Number-Letter Level)]/ Completion time (Number-Letter level) x 100% | [1.94 (Exercise level) – 1.75 (Number-Letter Level)]/ 1.75 (Number Letter Level) x 100 = **10.86** |
| Cognitive cost (moderate) | Cognitive | Number-Letter level (standard condition) Vs. Go-No-Go level (standard condition)* | [Completion time (Go-No-Go level) – Completion time (Number-Letter Level)]/ Completion time (Number-Letter level) x 100% | [1.74 (Go-No-Go level) – 1.65 (Number-Letter Level)]/ 1.65 (Number-Letter Level) x 100 = **5.45** |
| Cognitive cost (high) | Cognitive | Number-Letter level (standard condition) Vs. Stroop level (standard condition)* | [Completion time (Stroop level) – Completion time (Number-Letter level)]/ Completion time (Number-Letter level) x 100% | [2.06 (Stroop level) – 1.68 (Number-Letter level)]/ 1.68 (Number-Letter level) x 100 = **22.62** |
| Auditory interference cost | Perceptual | Average change between standard condition Vs. auditory interference condition across all levels | [Completion time (average of all auditory condition reps) – Completion time (average of standard condition reps for each level)]/ Completion time (average of standard condition reps for each level) x 100% | [1.81 (average of all auditory condition reps) – 1.66 (average of standard condition reps for each level)]/ 1.66 (average of standard condition reps for each level) x 100 = **9.04** |
| Scramble cost | Multi-domain switching | Average change between standard condition Vs. scramble condition across all levels | [Completion time (average of all scramble condition reps) – Completion time (average of standard condition reps for each level)]/ Completion time (average of standard condition reps for each level) x 100% | [1.44 (average of all scramble condition reps) – 1.18 (average of standard condition rep for each level)]/ 1.18 (average of standard condition rep for each level) x 100 = **22.03** |
| Fatigue cost | Fatigue | Motor task (pre) Vs. motor task (post)* | [Completion time (motor task post) – Completion time (motor task pre)]/ Completion time (motor task pre) x 100% | [0.93 (motor task post) – 0.83 (motor task pre)]/ 0.83 (motor task pre) x 100 = **12.05** |

Rep, repetition.

*Comparison uses the best performance of the two repetitions for the standard condition and motor task (i.e., shortest completion time, lowest average heart rate, fewest errors).

**Calculations also computed using errors and average heart rate as comparison metric.

***Example completion times reported using standardized metric of seconds per tablet button selection (seconds/tablet).

assessment. HR collection was integrated within the *R2Play* system, with average and maximum HR automatically computed for each level and repetition.

**Perceived exertion:** The ten-point Children's OMNI scale was embedded within the *R2Play* clinician interface to collect ratings of perceived exertion (RPE) from youth participants during check-ins after each level. The OMNI scale is a validated tool for measuring RPE among youth during exercis [31].

**Concussion-related symptoms.** Before beginning the *R2Play* assessment, youth completed the Post-Concussion Symptom Inventory (PCSI) within the *R2Play* clinician interface to rate the presence of any concussion-related symptoms across physical, cognitive, emotional and sleep/fatigue domains [32]. Between each *R2Play* level, clinicians administered a symptom check-in to assess any changes in the concussion-related symptoms from the PCSI, with the following response options: (1) "No symptoms", (2) "Some symptoms but did not get worse", (3) "Symptoms got worse", (4) "Symptoms so much worse I had to stop."

**System usability scale.** The SUS is a validated self-report tool for subjective assessment of technology usability based on 10 statements rated on a five-point Likert Scale from strongly disagree (1) to strongly agree (5) [33]. Each item is attributed a score contribution from 0-4 with the total SUS score out of 100 obtained by summing individual item scores and multiplying by 2.5. A total SUS score of 68 is deemed average, 68–80 is considered good, and ≥80 excellent. The *R2Play* software interface achieved an excellent usability score (SUS = 81 ± 8.02) from clinicians during cognitive walkthrough exercises as part of system development [28]. The current study is the first to evaluate usability of administering *R2Play* with youth participants.

**Semi-structured qualitative interviews.** Semi-structured interviews were conducted separately with clinician and youth participants after they finished the *R2Play* assessment and reviewed results together with a research team member. Youth interview questions focused on general impressions of *R2Play* and achievement of design objectives related to ease of use (e.g., "did you have a hard time learning the rules?), fun (e.g., "was *R2Play* fun?"), and sport-like (e.g., "did *R2Play* remind of you the skills you do in sports?"). Clinicians were asked to reflect on the informativeness of *R2Play* and its scoring measures (e.g., "do you think the assessment would be informative in RtoP decision making?"). Clinician follow-up interviews were conducted virtually via Zoom's videoconferencing platform and included questions targeting each design objective. Complete interview guides can be found in S2 Appendix. All interviews were audio recorded, transcribed, and checked for accuracy.

## Analysis

**Quantitative.** Participant demographics and baseline PCSI scores were summarized descriptively. The SUS was analyzed according to Brooke [34]. Mean total SUS scores (out of 100) were calculated separately for clinicians' first and second *R2Play* session to capture any changes in perceived usability upon repeat administration. The mean total SUS score was then compared to norm-referenced data, wherein a SUS score of 68 represents average usability [35], and assigned a letter grade based on the curved grading scale and percentile ranges developed by Sauro and Lewis [36].

*R2Play* results data was ascertained from automated system logs and analyzed using Microsoft Excel and R statistical software. Completion times, errors, and average HR for each level and condition were plotted to visually explore trends in relation to hypothesized level loading (**Table 2**). Multi-task cost scores were calculated using the equations in **Table 2** and summarized descriptively. Assessment duration was calculated as time between pre-assessment measures collection and final level completion. Physical exertion levels for each youth participant were established by calculating average and maximal HR across active assessment time (during repetition completion and within 10 seconds after repetition completion), expressed as a percentage of their age-predicated maximal HR based on the Tanaka equation ($208 - 0.7*age$), and highest reported RPE.

**Qualitative.** Clinician and youth interview transcripts were analyzed separately via content analysis using the design objectives of interest as a framework for organizing data [37]. Three researchers (JS, PG, DD) collaboratively coded most

data (two youth interviews, two clinician post-session interviews, and all five clinician follow-up interviews) to establish a preliminary codebook. Codes were derived inductively from the data without applying any theoretical framework to capture participants' original ideas, in their own words wherever possible. Codes derived from this initial review were transferred to NVivo qualitative analysis software and flexibly applied to remaining transcripts, allowing for the addition of new codes to capture emergent concepts. When new codes were added, consensus among team members was confirmed, and earlier transcripts were reviewed and re-coded as needed. This process was completed separately and independently by one researcher for clinician (JS) and youth (PG) transcripts, maintaining ongoing communication regarding codes and their interpretations. After all transcripts were coded, JS and PG separately organized the codes into categories constructed around the *R2Play* design goals. Categories and their interpretations were discussed and refined through three meetings with the tertiary reviewer (DD) and then finalized within the broader research team to establish study findings. Strategies to support qualitative trustworthiness included investigator triangulation through collaborative coding and discussion (described above), in-vivo coding (i.e., using participants' own words) wherever possible, and maintaining an audit trail of analysis.

## Results

Five clinicians and 10 healthy youth (ages 10–22 years) completed the study (**Table 3**). Within the sub-sections below, relevant quantitative and qualitative results are presented for each design objective from **Table 1**, followed by a brief description of key changes to the design of *R2Play* based on the results of this proof-of-concept study. A design table was created (**S3 Appendix**) to summarize participant suggestions in relation to design objectives, many of which are discussed in the sub-sections below, and map feedback onto subsequent design iterations. Given the pilot nature of this work and preliminary system prototype, minor technical issues caused challenges with data collection (e.g., extended assessment duration, incomplete HR data, missing level completion times). Such cases are clearly identified throughout the results. When encountered, technical problems were investigated via system logs, field notes, and video/screen recordings to

**Table 3. Clinician (C) and youth (Y) participant characteristics.**

| Participant ID | Age Range (Years) | Gender | Clinical Discipline | Practice Setting | Years of Experience | Client Age Range (Years) |
|---|---|---|---|---|---|---|
| C1 | 25-34 | F | Occupational Therapy | Public and private | 6 | 5-21 |
| C2 | 45-54 | M | Athletic Therapy | National Sports Organization | 20+ | 17+ |
| C3 | 25-34 | F | Occupational Therapy | Public and private | 2 | 0-18 |
| C4 | 35-44 | F | Medicine (Developmental Pediatrics) | Public and private | 6 | 0-19 |
| C5 | 25-34 | F | Physiotherapy | Public and private | 3.6 | 0-18 |

| Participant ID | Age (Years) | Gender | Sports Played (Hours Per Week) | | | |
|---|---|---|---|---|---|---|
| Y1 | 14 | M | Baseball (10); Swimming (2) | | | |
| Y2 | 14 | M | Hockey (4.5); Soccer (4) | | | |
| Y3 | 13 | M | Baseball (8) | | | |
| Y4 | 13 | F | Hockey (12); Lacrosse (12); Ball Hockey (2) | | | |
| Y5 | 15 | M | Snowboarding (8); Rock Climbing (2) | | | |
| Y6* | 21 | F | Wheelchair Basketball (15) | | | |
| Y7 | 22 | M | Hockey (3); Squash (2) | | | |
| Y8 | 15 | M | Volleyball (4–6) | | | |
| Y9 | 22 | F | Badminton (8–10) | | | |
| Y10 | 10 | F | Figure Skating (25); Gymnastics (3) | | | |

*Participant Y6 is a wheelchair user and completed *R2Play* in a wheelchair basketball chair.

identify causes and resolve issues between sessions to prevent re-occurrence and maximize data completeness. Records with incomplete data were removed from relevant analyses.

**Design objective 1: Ease of use**

**SUS.** The mean total SUS score was 76.5±5.76 (95% confidence interval [CI]: 71.5, 81.5) for clinicians' first *R2Play* assessment session and 81±8.4 (95% CI: 73.6, 88.4)) for their second *R2Play* assessment, potentially indicating a slight improvement in perceived usability with repeat administration. A SUS score of 81 places *R2Play* within the 90–95th percentile range of usability, corresponding to a grade of "A" for "good-to-excellent" usability [36]. Raw SUS scores for each participant can be found in S4 Appendix.

**Clinician qualitative feedback.** All clinicians described *R2Play* as easy to learn and use with little practice. After two sessions, clinicians felt comfortable using the system independently without technical support: "I did think it was quite easy to use… the second time it was a lot smoother because you knew what to expect. So, I think the learning curve for it is quite low, like you don't need to learn a lot to be able to use it." (C5). As detailed below, clinicians identified specific aspects of *R2Play* that were particularly usable or difficult to use and provided suggestions to improve usability.

*Usability successes:* Specific aspects of *R2Play* that clinicians identified as particularly usable included:

• Automatic prompts and instructions built into the interface screens.

• On-screen visuals to help communicate with youth.

• Time-stamped notes to capture clinical observations.

• Multiple methods for training participants on level instructions including options for visual and video demonstrations.

*Usability issues:* Features that clinicians identified as usability issues included screens that were difficult to navigate, buttons that could not be selected properly, and awkward wording of response options.

*Usability suggestions:* Clinicians provided the following suggestions to improve usability of *R2Play*:

• Provide a visual summary of the overall assessment protocol to help follow progress.

• Use a color-coded system to display heart rate based on ranges of exercise intensity.

• Allow notes for clinical observations to be written during breaks between repetitions.

• Begin breaks between repetitions automatically rather than requiring initiation.

**Youth qualitative feedback.** Youth perceived *R2Play* as highly usable since the instructions were easy to learn, training methods were effective, and levels progressed appropriately. They did not identify any usability issues or suggest any changes with respect to usability.

*Easy to learn:* There was unanimous agreement among youth that the tasks in *R2Play* were straightforward and easy to learn. For example, one youth said, "I liked that it was simple… there wasn't a huge process of trying to understand how it worked, and you could just get right into it… understanding the rules itself I think was pretty straightforward." (Y7).

*Effective training:* Youth reported that level instructions provided were clear and easy to understand. Training methods were believed to be effective for teaching level rules, with the instructional videos identified as most useful: "…the videos, they teach you exactly what to do." (Y1)

*Appropriate level progression:* Feedback from youth indicated that level difficulty progressed appropriately, allowing them to become comfortable with the *R2Play* system and baseline task before introducing new rules and challenges. There was an adjustment period at the start during which youth became familiar with task rules and how to interact with the system, after which they felt more comfortable. For example, "At first it was a little bit confusing, I—I wasn't familiar

with how to go, I was curious on like what would happen… but as I went along, I got familiar with it, it wasn't difficult, and I just like kind of knew what was gonna happen." (Y5).

### Design objective 2: Fun

**Clinician qualitative feedback.** Clinicians described how *R2Play* could be a uniquely fun assessment for youth because of its gamification and aspects of challenge and competition. However, tediousness and stress were identified as potential issues that may make *R2Play* less enjoyable for some youth.

*Gamification:* Clinicians viewed the gamified nature of *R2Play* as particularly relatable and enjoyable for youth: "We're in a video age, right? … [maybe] they could relate to it a lot better, and it would be a lot more fun than just if it's doing something on a piece of paper." (C2). Gamification was seen as a helpful strategy for establishing rapport with youth through physical activity.

*Challenge and competition:* The novelty, challenge, and scoring of the underlying trail making task and layered levels and conditions were seen as strengths of *R2Play* that could help keep youth more engaged, especially among athletes who enjoy competition. For example, "The added challenge I think makes it more fun… I think that's what we want, we want something that is going to push them, but then also something that they're going to be engaged [in] and keep trying to do their best at…" (C3).

*Potential tediousness and stress triggers:* Clinicians suggested that some youth, particularly those who are not competitive athletes, could find *R2Play* boring or tedious as it involves several repetitions without performance feedback. There was also some concern that youth who are anxious or fear avoidant after their injury may find *R2Play* stressful because of its complexity. Knowing the client well and having an established therapeutic relationship was therefore described as important before introducing *R2Play*. Performance feedback and positive reinforcement were also recommended to help ensure the participant's comfort and increase motivation.

**Youth qualitative feedback.** Youth reported that the baseline *R2Play* task felt like a game, which made the assessment fun. Some youth also indicated that they enjoyed being challenged cognitively: "I really enjoyed [*R2Play*], it was a lot of fun running around from [tablet] to [tablet]… it was fun being puzzled" (Y5). The challenge level of *R2Play* was seen as appropriate in that it was difficult enough to be engaging but not so difficult that it felt discouraging. Youth did not express any boredom from task repetition, or stress related to task complexity and performance pressure.

### Design objective 3: Sport-like

**Physical exertion data.** Table 4 presents the average and peak HR and peak RPE for each participant during the *R2Play* assessment. Overall, *R2Play* elicited moderate-to-vigorous physical exertion, with participants averaging approximately 60–85% of their age-predicted $HR_{max}$ across the entire assessment and achieving a peak of $83.1 \pm 9.2\%$ (95% CI: 77.4%, 88.5%) of their age-predicted $HR_{max}$. Highest reported (peak) RPE ranged from 2-7 (mean: $4 \pm 1.67$; 95% CI: 2.9, 5.1). Peak HR occurred in the exercise level for all participants, while peak RPE was reported following either the exercise or Stroop level. Full RPE results for each participant can be found in S5 Appendix.

**Clinician qualitative feedback.** Clinicians generally perceived *R2Play* to reflect the demands of sport well, with some viewing it as a significant transformation of a clinic setting: "this created a more demanding system than I would be able to create… definitely was the most realistic sport environment I've seen created in this room." (C1). The relevance of *R2Play* was thought to depend on specific activities to which the client is returning, with some clinicians suggesting it may be less applicable to individual sports than team sports. Clinicians also emphasized the lack of sport-specific skills in *R2Play*. Nevertheless, aspects of sport were noted across physical, cognitive, and perceptual domains.

*Physical conditioning:* *R2Play* was considered a good challenge of physical conditioning and motor functions including coordination and agility (e.g., acceleration, changes in direction). C3 summarized how *R2Play* addresses many relevant areas of physical performance, saying "When I look at physical components, I try to do a bit balance and

**Table 4. Average and peak HR and peak RPE for each participant.**

| Participant | Average HR | | Peak HR | | Peak RPE |
|---|---|---|---|---|---|
| | Raw | %Max | Raw | %Max | |
| Y1 | 154 | 77.7 | 184 | 92.8 | 5 |
| Y2 | 128 | 64.6 | 155 | 78.2 | 4 |
| Y3 | 172 | 86.5 | 196 | 98.5 | 7 |
| Y4 | 137 | 68.9 | 156 | 78.4 | 2 |
| Y5 | – | – | 155 | 78.5 | 3 |
| Y6 | 117 | 60.5 | 152 | 78.6 | 2 |
| Y7 | 155 | 80.5 | 184 | 95.5 | 5 |
| Y8 | 115 | 58.2 | 132 | 66.8 | 4 |
| Y9 | – | – | 158 | 82.0 | 2 |
| Y10 | 134 | 66.7 | 163 | 81.1 | 6 |
| Mean ± SD (95% CI) | 139.0 ± 19.9 (125.2, 152.8) | 70.4 ± 10.1 (63.5, 77.4) | 163.5 ± 19.0 (151.7, 175.3) | 83.1 ± 9.2 (77.4, 88.8) | 4 ± 1.67 (2.9, 5.1) |

HR, heart rate; %Max, percentage of age-predicted maximum heart rate; RPE, rating of perceived exertion; CI, confidence interval. Average heart rate could not be calculated for Y5 and Y9 due to technical issues within the system causing loss of synchronization between HR data and event markers (timestamps) for each level.

coordination and speed and strength, and I think that this assessment could cover all of those things… and then coordination definitely, and kind of switching direction quickly, so I like that component of it." (C3). Clinicians noted that the level of exertion achieved during *R2Play* varied between individual participants, with some reaching maximal capacity and others not, possibly due to differences in their approach towards the speed-accuracy trade-off. Opportunities for increased exertional demands were therefore recommended to ensure that athletes are tested to their full capacity.

***Cognitive demands:*** Clinicians appreciated that *R2Play* embedded cognitive challenges within a physically exerting assessment using tasks grounded in established neuropsychology testing paradigms (i.e., Trail Making, Stroop). This combined physical and cognitive challenge in *R2Play* was seen as a novel approach to match the demands of sport in clinic: "It's the closest I've ever been able to simulate a sport environment in a hospital setting… I've never been able to do anything that's that demanding cognitively, like doing a cognitive and physical combination together, so that was really great." (C1). Cognitive elements of sport that clinicians specifically recognized in *R2Play* included multi-tasking, rule maintenance, and spontaneous decision-making within a changing environment. The scramble condition was seen as particularly relevant to decision-making in sport: "I think [the scramble condition] really emulates a game. If the ball, or the puck, or something changes direction, and you have to… stop and figure out where to go." (C3).

***Perceptual skills:*** Clinicians noted areas of perceptual function that are addressed through participants' interactions with the environment during *R2Play*, including visual scanning, auditory processing, and vestibular function related to dynamic balance and positional changes of the head. The background noise condition was seen as particularly useful for identifying ongoing symptoms or changes in auditory function that are highly relevant to sport, for example, "I also like the idea of the noise interference a lot… kids who have a concussion that maybe were experiencing noise sensitivity, that might trigger some of their symptoms which might indicate, maybe holding back… most sports have some background noise." (C3). However, some clinicians suggested louder background noise or distinct audio clips for each level to make it harder to ignore.

**Youth qualitative feedback.** ***Dynamic physical exertion:*** Physical exertion and running were the most common elements of sport that youth identified in *R2Play,* with some referencing it as a "workout". Overall, some youth found *R2Play* moderately exerting, while others felt it was too easy, and the exercise level was consistently said to be most physically demanding. Multi-planar movement was an aspect of *R2Play* seen to be relevant to sport: "…in basketball you

want to try a layup, you would like go through certain movements left and right, which was somewhat like the going to one, two, then three, so it was definitely reminiscent of a sport like that" (Y5).

*Cognitive challenge:* While individual task rules were seen as simple and easy to understand, the layering and integration of tasks during *R2Play* challenged youth in a manner reminiscent of sport. Specific thinking skills that youth mentioned using during *R2Play* included rule maintenance, planning, and attentional control. Broadly, the need to apply these thinking skills while moving reminded youth of sport: "…focusing on the numbers and which one you were trying to tap… I think that's similar to trying to focus in hockey, like to make sure you have the puck, and lacrosse when you make sure you have the ball with a defender in front of you, so I think the numbers part kind of helped you focus on something else when you're moving" (Y4). As expected, most youth found the Stroop level most cognitively challenging, while others said the exercise level was most challenging because it involved more multi-tasking and required stimulus-response differentiation (i.e., responding appropriately to exercise tone versus scramble tone).

*Movement skills and agility:* Sport-related movement skills and agility were seen to be reflected in *R2Play*. Youth described how *R2Play* required them to lunge, shuffle, reach, change directions, and apply quick footwork in a manner similar to sport. Many referenced specific sport drills they saw as having movement demands similar to *R2Play,* for example, "It reminds me of footwork drills in badminton… we would call a scramble, you just point somewhere, and you have to go, but [in *R2Play*] it's more like—instead of pointing, it's like that letter or number is telling you where to go." (Y9).

**Design objective 4: Potential clinical value**

**Concussion-related symptoms.** The median baseline PCSI symptom score was 4 (range 0–11), which is consistent with normative data for youth athletes without concussion [38] As expected, participants reported "no symptoms" or "some symptoms but did not get worse" for all post-level symptom check-ins, indicating no change in concussion-related symptoms throughout the course of *R2Play*. Baseline PCSI scores and symptom check-in responses for each participant can be found in S6 Appendix.

**R2Play assessment results.** Raw *R2Play* results for level completion times (in seconds/button) and HR are shown in **Fig 4**. Results are presented for n = 9 as data was missing for Y9 due to a technical issue in automated system output. Across levels in the standard condition, completion time was generally highest in the Stroop level (high cognitive load) compared to other levels (panel A), while HR was typically elevated in the exercise and Stroop levels (panel B). Within levels, completion time and HR were slightly higher in the auditory condition (perceptual load) compared to the standard condition (panels C and D) and consistently elevated in the scramble condition (multi-domain switching) compared to standard condition (panels E and F).

**Table 5** presents the *R2Play* multi-task cost score results for completion time and heart rate. For *completion time*, the largest performance costs were seen in the cognitive (high) and scramble costs, followed by the auditory interference cost, while other *completion time* cost scores did not demonstrate consistent alignment with hypothesized level loading (Table 2). All *HR* cost score results aligned with hypothesized level loading, with the greatest HR performance costs occurring in the fatigue cost, cognitive cost (high) and exertion costs. Error-based cost scores could not be calculated due to lack of errors across multiple levels and conditions (S7 Appendix). Cost score results from each participant can be found in S8 Appendix.

**Clinician qualitative feedback.** Clinician feedback highlighted the potential value of *R2Play* for assessing sport tolerance and enabling rich clinical observations, while a need for further research and knowledge translation was identified to support the use of *R2Play* cost score metrics in practice. In addition to a RtoP assessment tool, clinicians also saw value in *R2Play* as a therapeutic rehabilitation tool.

*Assessing sport tolerance:* Clinicians felt that *R2Play* would be most useful for assessing clients' tolerance to a spor environment. For example, C1 said, "If I were to use it, it would be like I really need to challenge this kid, I want to see how they're able to tolerate this activity, I don't really have a good idea of what they're able to do, or if they can basically tolerate a sport-like environment." For this purpose, clinicians said they would primarily look for changes in

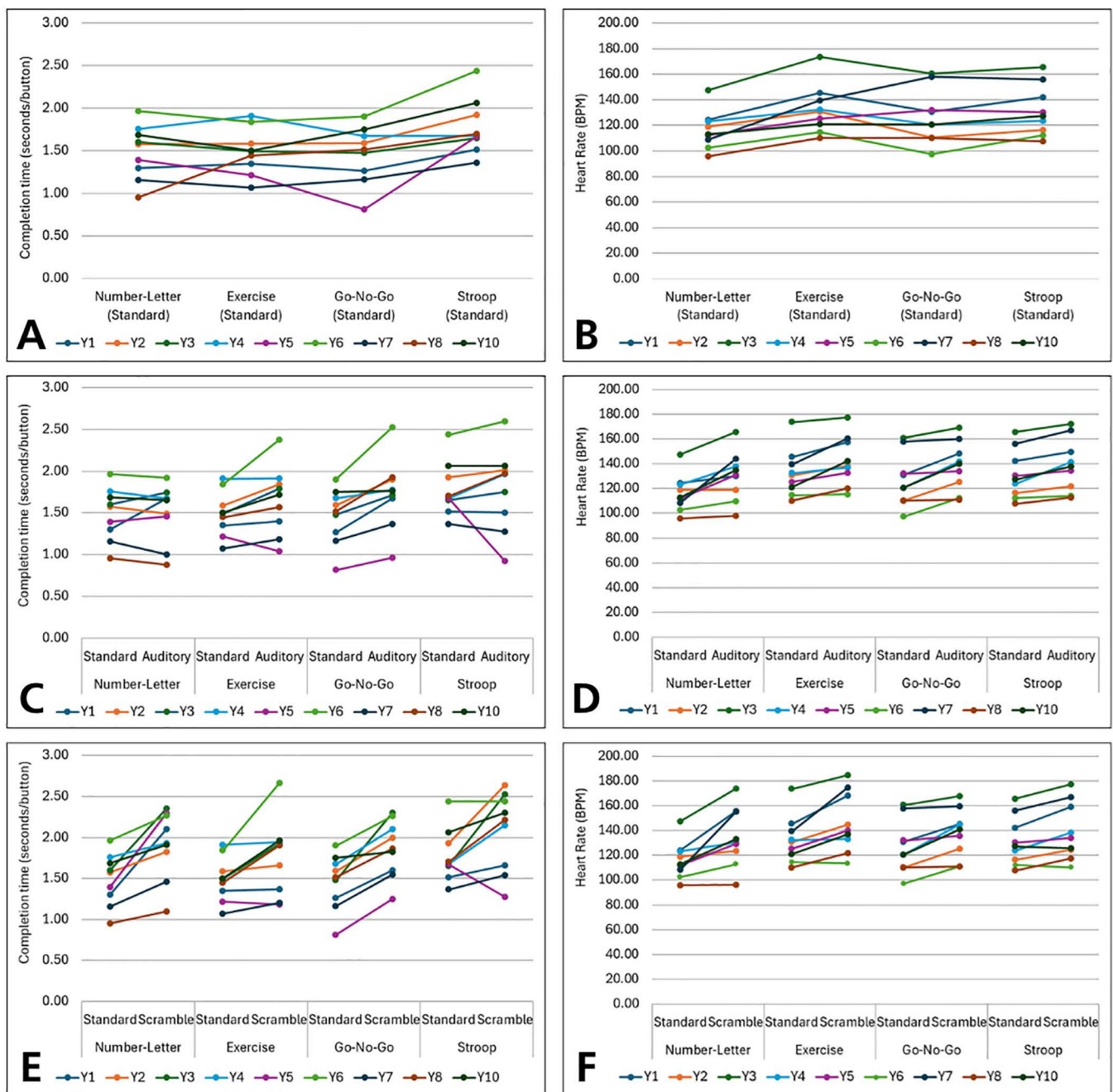

**Fig 4. Raw *R2Play* results for each participant (n = 9) showing changes in performance across levels and conditions.** Completion time (A) and average HR (B) for standard condition of each level; Completion time (C) and average HR (D) for auditory and standard condition of each level; Completion time (E) and average HR (F) for scramble and standard condition of each level. For standard condition data, the best of two repetitions is reported (i.e., lowest completion time or average HR). *Note.* Participant Y6 is a wheelchair user.

concussion-related symptoms during the test to indicate whether an athlete was ready for RtoP. Physical endurance was another component of sport tolerance that clinicians appreciated being able to evaluate through *R2Play* by monitoring HR and RPE.

**Table 5.** *R2Play* cost score results for completion time and average heart rate (n = 9).

| | Completion time | Heart rate |
|---|---|---|
| Exertion cost | -6.52 (-7.69-2.82) | 11.71 (9.88-16.97) |
| Cognitive cost (moderate) | -2.88 (-4.74-0.69) | 6.71 (-2.02-15.07) |
| Cognitive cost (high) | 20.00 (16.35-22.45) | 12.42 (9.45-14.41) |
| Fatigue cost | -11.63 (-17.82-10.57) | 24.62 (13.16-27.99) |
| Auditory interference cost | 8.74 (3.27-12.76) | 6.94 (6.02-14.06) |
| Scramble cost | 21.09 (20.74-24.98) | 9.04 (8.21-11.77) |

Cost scores are expressed as the percentage change in performance between levels or conditions (*see formulas in* Table 2). Data presented as median (IQR). Calculations used the best of the two repetitions for motor trails and standard condition (i.e., lowest completion time or average HR).

*Enabling rich clinical observations:* *R2Play* was thought to enable rich clinical observations regarding aspects of readiness for RtoP including dynamic balance control, coordination, concentration, and decision making within a complex simulated sport environment. For example, in describing how she might use *R2Play*, C3 said "I would look at coordination… if someone was having a lot of trouble kind of re-focusing, especially during the scramble, or if they were tripping, or any kind of appearance of struggle, like holding your head or grimacing or being super out of breath." (C3). Through these observations, *R2Play* was seen to possibly help bridge the gap between clinical assessment and performance in sport, a common limitation in the current approach for RtoP clearance:

> "…when I'm clearing someone, it's based on symptoms and a physical exam, and it's very subjective… they'll tell you 'Oh, I went to practice, and this went well, and the coach was happy', but you don't actually see how they're functioning, right? So having different stimuli, having a physical component, a cognitive component, it's more comprehensive than what I would be able to do in clinic, and hits a lot of the different things that can trigger symptoms, or tip you off that someone might not be quite ready." (C4).

*Cost score metrics:* There was a shared sentiment among clinicians that the *R2Play* multi-task cost score metrics, while interesting, could be difficult to interpret and apply in practice. For C3, this related to the volume and presentation of data: "I think the results are really interesting… it's great to have lots, but I think right now I was a little bit overwhelmed with all of the different things, and I liked that there's like a snapshot of it but some of the things are hard to interpret." (C3). The need for support with results interpretation was also highlighted by C1, saying "I feel like I'd have to learn from you guys on how to make sense of these." Validated age-based normative data was a direction of interest for some clinicians to help interpret these metrics. Concerns were also raised regarding learning effects related to serial assessment and repeat concussions, requiring further investigation. Ultimately, clinicians believed that cost scores should be interpreted within the overall context of the individual and supported by clinical judgement, avoiding definitive RtoP decisions based solely on these metrics.

*Concussion rehabilitation tool:* Clinicians highlighted the potential utility of *R2Play* as a therapeutic concussion rehabilitation tool. For example, it could help facilitate active rehabilitation: "I would use [*R2Play*] mostly as a rehab tool to encourage more exercise, because even if someone doesn't have a sport goal, doing exercise is going to help their recovery, and if they're not doing exercise… then we have to find a way to get them to do it… sometimes you actually have to get them to physically do it and see the benefit for themselves." (C1). By demonstrating successful experiences in complex activity through progressive level achievement, *R2Play* was proposed as a strategy to address anxiety and fear avoidance behaviour among youth with concussion and their parents.

## Design objective 5: Resource efficiency

**Assessment duration.**  The median total duration of *R2Play* assessments was 32.22 minutes (IQR = 4.32; range = 27.79-46.86). One session was impacted by technical issues (unresponsive equipment) causing substantially extended duration. **Table 6** presents the duration of *R2Play* assessments for each session, broken down by configuration time (assessment initiation, customization, baseline measures collection) and active assessment time (i.e., completing levels).

**System cost.**  The *R2Play* system prototype was built using entirely publicly available commercial equipment, with a total hardware cost of approximately $4,000. S9 Appendix outlines the individual equipment components and associated costs.

**Clinician qualitative feedback.**  *Time:*  There was broad consensus among clinicians that the current length of the *R2Play* assessment and time required for set-up, history taking, customization, rest breaks, and results review, amounting to 30–40 minutes total, could be a barrier to clinical implementation. Comments suggested that the practicality of a shorter assessment must be balanced against the need for prolonged exertion to adequately fatigue participants in a manner similar to sport: "How long is a practice? It's going to be at least two hours to three hours and so… anything under 20 minutes I think would definitely be too short in terms of getting the data quality that you want." (C2). The ideal assessment duration was believed to be 20–30 minutes. Suggestions for reducing assessment duration included removing the Go-No-Go or Stroop level, reducing the number of repetitions, and shortening rest breaks.

*Cost:*  The cost of the *R2Play* system equipment was described as a potential barrier, depending on clinicians' practice setting (i.e., public vs. private), client population (e.g., community, high-performance athletics) and model of care (e.g., fee-for-service, consultative). One clinician stated that the cost of *R2Play* should be justified by demonstrating its value over traditional measures, saying "How different is this than like a standardized shuttle test? I know there's a lot of really good information that's interesting to me and that I'd want to see [in *R2Play*], but I guess I'd want to compare the two and see if it's actually that much more worth it to go with the more expensive technological option." (C5). Cost discussions also emphasized bundled packages for price inclusions (e.g., equipment, software, training materials, consultation on results interpretation), compatibility with different devices and operating systems, and integration with existing organizational equipment as strategies to help manage costs.

*Personnel and expertise:*  Resource considerations also related to personnel and expertise, specifically which healthcare professionals would use *R2Play,* and how they would be trained. Clinicians noted that while the

**Table 6.**  *R2Play* assessment duration (minutes) by configuration and active assessment time.

| Session number | Configuration | Active assessment | Total |
|---|---|---|---|
| 1 | 5.38 | 41.48* | 46.86 |
| 2 | 6.05 | 25.49 | 31.54 |
| 3 | 4.42 | 27.32 | 31.74 |
| 4 | 7.53 | 25.16 | 32.69 |
| 5 | 3.46 | 24.33 | 27.79 |
| 6 | 5.97 | 23.27 | 29.24 |
| 7 | 6.34 | 29.58 | 35.92 |
| 8 | 5.76 | 23.27 | 29.03 |
| 9 | 6.16 | 28.44 | 34.60 |
| 10 | 6.93 | 24.25 | 31.18 |
| Median (IQR) | **6.01 (0.82)** | **28.41 (4.76)** | **32.22 (4.32)** |

*Assessment interrupted by technical issues (unresponsive equipment).

responsibility to provide clearance for RtoP is restricted to physicians and nurse practitioners in many jurisdictions, these professionals often would not have time or space to use *R2Play*. One clinician suggested that allied health professionals could administer *R2Play* and share results with medical providers: "You don't need medical expertise to run [*R2Play*]… the physio might run it, or the nurse might run it, or the [occupational therapist] might run it… and then just have the physician review the results as part of the medical clearance." (C4). To support implementation, clinician feedback indicated a need for training and knowledge translation resources, particularly focusing on interpretation of results for decision-making.

### Design objective 6: Flexibility

**Clinician qualitative feedback.** *Space, equipment, and set-up:* Some clinicians described concerns about having adequate space for *R2Play* within their practice settings. High-performance athletics, sport medicine, and concussion specialty clinics were seen as appropriate settings that may have suitable space. The number of equipment components and moving parts within the *R2Play* system was identified as a potential challenge by introducing risk for technical issues, such as devices not being charged or communicating properly. Setting up the system within its standardized configuration was also seen as a challenge, which could amplify time constraints, and clinicians recommended streamlining this process. In thinking about implementation, some clinicians believed that collegiate or professional sport teams would be a large commercial market that may be best equipped with resources to support *R2Play*.

*Appropriateness for individual clients:* Reflecting on their practice, clinicians felt that *R2Play* would be appropriate for some, but not all clients. Age was identified as an important factor, as the complexity of *R2Play* may be unsuitable for younger children. Some clinicians also spoke of differences in the RtoP process for younger children making *R2Play* less applicable: "I don't see the purpose in a young child because they're not playing contact sports, they're not doing as high-risk stuff." (C4). Given the time required for *R2Play*, clinicians emphasized that it must align with clients' goals. Competing priorities such as school often require attention and could limit time available for *R2Play.*

*Adapting to individual client needs:* Clinicians thought that *R2Play* could be adapted for younger children by simplifying the task but felt it may be less applicable for a younger population due to their goals and activities. Adaptation suggestions for younger children included shortening the trail length, using only numbers or letters, incorporating pictures or symbols, and using developmentally appropriate symptom scales. Three clinicians also discussed how *R2Play* could be applied to adults without significant adaptation.

*Expansion to other clinical populations:* Two clinicians expressed an interest in using *R2Play* among youth with other forms of brain injury. They proposed that *R2Play* could offer a safe sport-like activity for youth with moderate-severe traumatic brain injury before being cleared for higher-risk activities, or could be used to test balance and coordination. Suggested modifications to accommodate youth with more severe brain injuries included shortening overall assessment duration and lengthening breaks. *R2Play* was viewed as a transferrable assessment applicable for a variety of other clinical populations: "People with various other medical conditions would be another thing to think about as well, because really what you're testing is agility and cognition and like physical conditioning, which is helpful in so many other conditions too." (C4). For example, one clinician suggested using *R2Play* as an assessment tool for athletes with musculoskeletal injuries to evaluate movement quality and footwork in the context of a physical-cognitive dual-task environment, while another described its use as a useful functional assessment for people with neuromuscular conditions.

### *R2Play* design iteration

Several changes were made to the design of *R2Play* based on findings from this study. **Table 7** summarizes key findings regarding major areas for improvement with respect to the *R2Play* design objectives and subsequent changes that were implemented in the design of *R2Play*.

**Table 7. Sample key findings informing changes to *R2Play* system design and research protocol.**

| Objective | Key findings | Supporting evidence | Design and research changes |
|---|---|---|---|
| Ease of use | Minor usability issues | • Clinician feedback (category: *Usability issues*) | • Re-designed PCSI screen for easier navigation<br>• Updated symptom check-in response options |
| | Usability suggestions | • Clinician feedback (category: *Usability suggestions*) | • Added "map" of assessment protocol to clinician interface<br>• Implemented colour-coded system to display heart rate in clinician interface based on exercise intensity ranges<br>• Enabled clinical notes function during breaks between repetitions<br>• Set automatic start for breaks |
| Fun for youth athletes | Potentially stressful for youth who are very anxious or fear avoidant | • Clinician feedback (category: *Tediousness and stress*) | • Added Injury Psychological Readiness to Return to Sport Scale[39] as a measure within subsequent studies<br>• Added new interview questions to explore anxiety and fear avoidance in *R2Play* in subsequent studies |
| Sport-like | Variable physical exertion | • Exertion data (Table 4)<br>• Exertion cost (Table 5)<br>• Clinician and youth feedback (categories: *Physical conditioning* and *Dynamic physical exertion*) | • Standardized exercise task in exercise level (burpees)<br>• Removed Go-No-Go level to streamline exertion<br>• Changed motor trail to "sprint check" requiring all-out sprint |
| Potential clinical value | Lack of useful information from Go-No-Go level | • Negative value for cognitive cost score – moderate (Table 5) | • Removed Go-No-Go level |
| | Insufficient cognitive challenge | • Infrequent errors across all levels | • Increased cognitive challenge by having trail continue upwards after first six nodes completed (…4-D-5-E-6-F)<br>• Added cognitive RPE rating to check-in between levels |
| Resource efficiency | Time as potential barrier to clinical implementation | • Assessment duration data (Table 6)<br>• Clinician feedback (category: *Time*) | • Removed Go-No-Go level to streamline assessment<br>• Set rest breaks between repetitions to start automatically |
| | Training needed to administer and interpret *R2Play* | • Clinician feedback (category: *Personnel and expertise*) | • Developed comprehensive *R2Play* clinician training program for use in future studies |
| Flexibility | Challenge of precise system set-up | • Clinician feedback (Category: *Space, equipment, and set-up*) | • Streamlined *R2Play* system equipment set-up process<br>• Developed set-up guides for *R2Play* training program |

## Discussion

This study sought to establish proof-of-concept for *R2Play*, a multi-domain assessment tool for RtoP decision making among youth with concussion, by demonstrating alignment with initial design objectives (easy to use, fun, sport-like, informative, resource efficient, and flexible). Overall, findings supported proof of concept, with excellent usability, promising exertion potential, and positive feedback from youth and clinicians regarding these design objectives. *R2Play* results and multi-domain cost score analyses reflected some, but not all, aspects of hypothesized level demand loading. Areas for improvement were identified and addressed through documented iteration of *R2Play* design.

The *R2Play* paradigm is unique in its goal to replicate the demands of sport through a focus on integration across functional domains (i.e., cognitive, motor, perceptual, socio-emotional), dynamic multi-planar movement requiring whole-body coordination, and complex decision making involving quick reactions to spontaneous stimuli, all performed at high speed [23,28]. This initial study provides evidence of these aspects, with further changes implemented to optimize physical and cognitive demands. From our preliminary analyses and qualitative feedback, the Stroop level and scramble condition appear most robust and reflective of real-world sport demands. Given that concussion affects cognitive inhibition [40] and perception-action integration [18,20], which are targets of the Stroop level and scramble condition, respectively [28], the cognitive (high) and scramble costs may be promising candidates for future validation.

Interestingly, participants did not comment on socio-emotional aspects of sport reflected in *R2Play*, despite being considered a component of the sport-like objective (**Table 1**). As many athletes report fear, anxiety, or lack of confidence despite medical clearance to RtoP [41–43], it is worth exploring the potential for *R2Play* and other simulated sport assessment tools to support psychological readiness to RtoP. Specifically, our findings suggest that gamified tasks involving complex challenges and competitiveness (e.g., scoring) could help introduce performance stress within controlled clinical environments, which may be a useful approach for building athletes' confidence during the RtoP process [41–43] or addressing anxiety and fear avoidance that are common among youth with concussion [44,45] This is consistent with other literature demonstrating the benefits of gamification in rehabilitation on anxiety, fear, and self-efficacy in chronic pain populations [46].

This study offers insight regarding clinician perspectives on the value of novel RtoP assessment approaches for concussion. Currently, RtoP progression is primarily guided by relatively simple single-domain clinical assessments and athletes' self-reported responses to each stage of the graded RtoP protocol, often completed without direct clinician supervision [3,14]. Despite its importance for RtoP decision making, [47–49] multi-domain skill integration can be difficult to assess within the confines of clinical environments due to limited space and equipment, meaning that clinicians often rely on creativity and experience to devise appropriately challenging integration tasks [50]. *R2Play* adds to a growing body of increasingly complex assessment paradigms designed to improve recovery monitoring and RtoP decisions through tasks that better reflect the multifaceted demands of sport, including dynamic exertion testing [51–53],cognitive-motor dual-tasks [16,17], and perception-action integration [18,20]. Our qualitative findings indicate that clinicians value these novel assessment approaches as opportunities to gather unique observations related to RtoP readiness. Specifically, *R2Play* was appreciated for its potential to directly assess dynamic exertion tolerance through symptom and HR responses, and enable rich observations of movement quality (e.g., coordination, agility), cognitive performance (e.g., multi-tasking, decision making), and psychosocial reactions (e.g., stress, anxiety, frustration) in the context of a sport-like task. It is hoped that dynamic multi-domain assessments like *R2Play* will help clinicians identify subtle performance changes that increase risk for re-injury [24,48,49,54] and inform athletes' progression to higher-risk sport activities (i.e., stages 4–6 of graduated RtoP strategy).[3] As noted by clinicians in this study, multi-domain tools may also be useful to track longitudinal recovery and inform targeted active rehabilitation strategies that help address fear avoidance and facilitate complete recovery [55,56].

## Limitations and future directions

This study was a preliminary proof-of-concept evaluation to guide iterative refinement of *R2Play* and does not provide evidence for the efficacy of this novel assessment paradigm. While healthy youth were invited here to rapidly test and gather feedback on the initial prototype, findings from this uninjured sample may not translate directly to clinical utility. Future work involving youth with concussion is imperative to validate the *R2Play* assessment and understand the value of this novel assessment approach. We did not collect participants' socioeconomic, racial/ethnic, or geographic information, which are relevant factors for concussion incidence, care access, and engagement with digital health tools that should be explored in future work. Furthermore, although our clinician sample represents multiple rehabilitation professionals, generalizability to other clinical or sports medicine populations may be limited.

The level structure of *R2Play* and its physical, cognitive, and sensory loading require validation, as do the psychometric properties of multi-domain cost scores. Comprehensive training resources are needed to support *R2Play* administration and interpretation of cost score results, which should be developed in consultation with clinicians and evaluated across different clinical environments. Practical considerations for using *R2Play* in practice were also raised (time, cost, personnel/expertise)*,* warranting attention to assess the feasibility of *R2Play* across different clinical settings, and cost comparisons to existing approaches. While this study employed a unique multi-method approach involving both quantitative and qualitative data pertaining to *R2Play* design objectives, more deliberate integration and triangulation across methods in

future work may contribute to richer findings and a deeper understanding of the complex issues inherent to feasibility and psychometric validity [57,58]. Alongside ongoing research, regulatory pathways must be explored to enable broader clinical use (e.g., medical device licensing).

## Conclusion

This study provides proof of concept for *R2Play*, a new multi-domain RtoP assessment tool for youth with concussion, and informed its iterative refinement through design changes prior to evaluation among youth with concussion. Findings indicate that clinicians value novel multi-domain assessment approaches for concussion to gather rich insights regarding readiness to return to play, including dynamic exertion tolerance, movement quality, cognitive performance, and psychosocial reactions, suggesting a potential role for these tools in clinical practice.

This work exemplifies how creative applications of low-cost digital technologies can stimulate clinical innovation and help create more dynamic assessment tasks within clinical environments to aid in return-to-play decision making. It also demonstrates the value of small-scale iterative testing and end-user engagement as key steps in developing digital health technologies and novel assessment tools before embarking on broader evaluation and implementation.

## Supporting information

**S1 Appendix.** *R2Play* **Assessment Overview Video.** Caption: S1 Appendix provides a video overview of the *R2Play* assessment protocol. Each assessment level and repetition condition is described and demonstrated.
(MP4)

**S2 Appendix. Interview Guides.** Caption: S2 Appendix provides the interview guides used for this work. The youth interview guide appears first, followed by the clinician post-assessment interview guide, and then the clinician follow-up interview guide.
(PDF)

**S3 Appendix. Design Change Table.** S3 Appendix presents a design change table that was constructed to summarize participant suggestions in relation to the design objectives, and map feedback onto subsequent design iterations. Design changes suggestions from clinicians are presented first, followed by design change suggestions from youth.
(PDF)

**S4 Appendix. System Usability Scale Results** . S4 Appendix presents the raw System Usability Scale results from each clinician participant.
(XLSX)

**S5 Appendix. RPE Results.** S5 Appendix presents the raw RPE results from every level for each youth participant.
(PDF)

**S6 Appendix. PCSI and Symptom Check-in Scores.** S6 Appendix provides the baseline PCSI scores for each youth participant, as well as their symptom check-in response for each level.
(PDF)

**S7 Appendix.** *R2Play* **Error Results.** S7 Appendix presents the raw *R2Play* error results from every level for each youth participant.
(PDF)

**S8 Appendix.** *R2Play* **Cost Score Results.** S8 Appendix presents the raw *R2Play* cost score results from each youth participant.
(XLSX)

 21 / 24

**S9 Appendix.** *R2Play* **Equipment and Associated Costs.** S9 Appendix outlines the *R2Play* system equipment and costs. Equipment components are labeled on an image of the system and described further in a table below, including their purpose and cost.
(PDF)

## Acknowledgments

The authors wish to thank participants for their time and dedication to the study. We would like to acknowledge the efforts of members of the NOvEL Lab and PEARL Lab Teams (Bloorview Research Institute), specifically Brendan Lam, Hiba Al-Hakeem, and Ajmal Khan. We also thank the Holland Bloorview Concussion Clinic team for supporting our work.

## Author contributions

**Conceptualization:** Josh Shore, Danielle DuPlessis, Emily Lam, Elaine Biddiss, Shannon E. Scratch.

**Data curation:** Josh Shore, Pavreet Gill, Danielle DuPlessis, Fanny Hotzé.

**Formal analysis:** Josh Shore, Pavreet Gill, Danielle DuPlessis, Andrew Lovell.

**Funding acquisition:** Danielle DuPlessis, Andrea Hickling, Emily Lam, Elaine Biddiss, Shannon E. Scratch.

**Investigation:** Danielle DuPlessis, Emily Lam.

**Methodology:** Josh Shore, Pavreet Gill, Danielle DuPlessis, Emily Lam, Elaine Biddiss, Shannon E. Scratch.

**Project administration:** Andrew Lovell, Andrea Hickling, Elaine Biddiss, Shannon E. Scratch.

**Resources:** Elaine Biddiss, Shannon E. Scratch.

**Software:** Emily Lam, Fanny Hotzé.

**Supervision:** Andrea Hickling, Elaine Biddiss, Shannon E. Scratch.

**Validation:** Pavreet Gill, Andrew Lovell.

**Visualization:** Josh Shore, Pavreet Gill.

**Writing – original draft:** Josh Shore.

**Writing – review & editing:** Josh Shore, Pavreet Gill, Danielle DuPlessis, Andrew Lovell, Andrea Hickling, Emily Lam, Fanny Hotzé, Elaine Biddiss, Shannon E. Scratch.

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
