## [Decision Letter · Decision Letter 0]

5 Jun 2025

PDIG-D-25-00163Multi-method proof-of-concept evaluation for R2Play: a novel multi-domain return-to-play assessment tool for concussion.PLOS Digital Health Dear Dr. Shore, Thank you for submitting your manuscript to PLOS Digital Health. After careful consideration, we feel that it has merit but does not fully meet PLOS Digital Health's publication criteria as it currently stands. Therefore, we invite you to submit a revised version of the manuscript that addresses the points raised during the review process. Please submit your revised manuscript within 30 days Jul 05 2025 11:59PM. If you will need more time than this to complete your revisions, please reply to this message or contact the journal office at digitalhealth@plos.org.  Please include the following items when submitting your revised manuscript:* A rebuttal letter that responds to each point raised by the editor and reviewer(s). You should upload this letter as a separate file labeled 'Response to Reviewers '. This file does not need to include responses to any formatting updates and technical items listed in the 'Journal Requirements' section below.* A marked-up copy of your manuscript that highlights changes made to the original version. You should upload this as a separate file labeled 'Revised Manuscript with Track Changes '.* An unmarked version of your revised paper without tracked changes. You should upload this as a separate file labeled 'Manuscript '. If you would like to make changes to your financial disclosure, competing interests statement, or data availability statement, please make these updates within the submission form at the time of resubmission. Guidelines for resubmitting your figure files are available below the reviewer comments at the end of this letter. We look forward to receiving your revised manuscript. Kind regards, Shannon Freeman, PhDAcademic EditorPLOS Digital Health Shannon FreemanAcademic EditorPLOS Digital Health Leo Anthony CeliEditor-in-ChiefPLOS Digital Healthorcid.org/0000-0001-6712-6626 **Additional Editor Comments (if provided):**  Thank you for your submission to PLoS Digital Health. The reviewers have provided constructive feedback which the authors are encouraged to pay close attention to. Both reviewers saw great merit in this paper and recognize the value that it will have to the existing knowledge in this area. I look forward to receiving the revised manuscript.

Sincerely,

Dr. Shannon Freeman**Reviewers' Comments:**  Reviewer's Responses to Questions

**Comments to the Author**

1. Does this manuscript meet PLOS Digital Health’s publication criteria ? Is the manuscript technically sound, and do the data support the conclusions? The manuscript must describe methodologically and ethically rigorous research with conclusions that are appropriately drawn based on the data presented.

Reviewer #1: Yes

Reviewer #2: Yes

2. Has the statistical analysis been performed appropriately and rigorously?

Reviewer #1: I don't know

Reviewer #2: Yes

3. Have the authors made all data underlying the findings in their manuscript fully available (please refer to the Data Availability Statement at the start of the manuscript PDF file)?

Reviewer #1: Yes

Reviewer #2: Yes

4. Is the manuscript presented in an intelligible fashion and written in standard English?

Reviewer #1: Yes

Reviewer #2: Yes

5. Review Comments to the Author

Reviewer #1: This paper presents a proof-of-concept evaluation of R2Play, a novel multi-domain assessment tool designed to support return-to-play decision-making for youth with concussion. The authors used a multi-method approach to evaluate whether the R2Play prototype aligned with six design objectives: being easy to use, fun, sport-like, clinically valuable, resource efficient, and flexible. Five clinicians and ten healthy youth (ages 10-22) participated in testing sessions followed by semi-structured interviews. Quantitative measures included the System Usability Scale, heart rate monitoring, ratings of perceived exertion, completion times, errors, and multi-task cost scores. Results showed that R2Play demonstrated good-to-excellent usability (SUS=81±8.4), achieved moderate-to-high intensity exertion (peak HR=80±11% age-predicted maximal), and was perceived as engaging and sport-like by participants. Clinicians valued R2Play's potential for observing multi-domain skill integration relevant to return-to-play readiness, though they noted concerns about resource requirements and result interpretation. The authors documented iterative design improvements based on study findings, including streamlining the assessment by removing redundant levels, standardizing physical exercises, and enhancing the user interface. This study supports proof-of-concept for R2Play as a promising assessment tool while acknowledging the need for further validation with youth with concussion.

Reviewer #2: Thank you for the opportunity to review this thoughtful and well-executed study titled "Multi-method proof-of-concept evaluation for R2Play: a novel multi-domain return-to-play assessment tool for concussion." The manuscript addresses a clear and timely need in pediatric and youth concussion assessment, particularly in bridging the gap between clinical testing and sport-specific demands.

Overall, the manuscript is clearly written and methodologically sound. The mixed-methods approach is appropriate for a proof-of-concept study and offers valuable insights into both system usability and the potential clinical value of R2Play. The qualitative and quantitative integration is effective and provides a comprehensive evaluation of the prototype. Attached is detailed comment aimed at enhancing clarity, accessibility, and utility for both research and clinical audiences.

6. PLOS authors have the option to publish the peer review history of their article (what does this mean? ). If published, this will include your full peer review and any attached files.

**Do you want your identity to be public for this peer review?** For information about this choice, including consent withdrawal, please see our Privacy Policy .

Reviewer #1: No

Reviewer #2: **Yes: ** ISAAC KWESI ACQUAH

---

## [Decision Letter · Decision Letter 1]

4 Sep 2025

PDIG-D-25-00163R1Multi-method proof-of-concept evaluation for R2Play: a novel multi-domain return-to-play assessment tool for concussion.PLOS Digital Health Dear Dr. Shore, Thank you for submitting your manuscript to PLOS Digital Health. After careful consideration, we feel that it has merit but does not fully meet PLOS Digital Health's publication criteria as it currently stands. Therefore, we invite you to submit a revised version of the manuscript that addresses the points raised during the review process. Please submit your revised manuscript within 30 days Oct 04 2025 11:59PM. If you will need more time than this to complete your revisions, please reply to this message or contact the journal office at digitalhealth@plos.org.  Please include the following items when submitting your revised manuscript:* A rebuttal letter that responds to each point raised by the editor and reviewer(s). You should upload this letter as a separate file labeled 'Response to Reviewers '. This file does not need to include responses to any formatting updates and technical items listed in the 'Journal Requirements' section below.* A marked-up copy of your manuscript that highlights changes made to the original version. You should upload this as a separate file labeled 'Revised Manuscript with Track Changes '.* An unmarked version of your revised paper without tracked changes. You should upload this as a separate file labeled 'Manuscript '. If you would like to make changes to your financial disclosure, competing interests statement, or data availability statement, please make these updates within the submission form at the time of resubmission. Guidelines for resubmitting your figure files are available below the reviewer comments at the end of this letter. We look forward to receiving your revised manuscript. Kind regards, Shannon Freeman, PhDAcademic EditorPLOS Digital Health Shannon FreemanAcademic EditorPLOS Digital Health Leo Anthony CeliEditor-in-ChiefPLOS Digital Healthorcid.org/0000-0001-6712-6626  **Journal Requirements:**  If the reviewer comments include a recommendation to cite specific previously published works, please review and evaluate these publications to determine whether they are relevant and should be cited. There is no requirement to cite these works unless the editor has indicated otherwise.  **Additional Editor Comments (if provided):**  Thank you so much for your patience in awaiting the reviewers feedback. As you can imagine, it can be difficult to recruit reviewers during the summer months. There are some very minor edits suggested by the reviewer and I concur. After addressing these few outstanding comments, I believe your paper will be strengthened and of great interest to a global audience.**Reviewers' Comments:**  Reviewer's Responses to Questions

**Comments to the Author**

1. If the authors have adequately addressed your comments raised in a previous round of review and you feel that this manuscript is now acceptable for publication, you may indicate that here to bypass the “Comments to the Author” section, enter your conflict of interest statement in the “Confidential to Editor” section, and submit your "Accept" recommendation.

Reviewer #3: All comments have been addressed

2. Does this manuscript meet PLOS Digital Health’s publication criteria ? Is the manuscript technically sound, and do the data support the conclusions? The manuscript must describe methodologically and ethically rigorous research with conclusions that are appropriately drawn based on the data presented.

Reviewer #3: Yes

3. Has the statistical analysis been performed appropriately and rigorously?

Reviewer #3: No

4. Have the authors made all data underlying the findings in their manuscript fully available (please refer to the Data Availability Statement at the start of the manuscript PDF file)?

Reviewer #3: Yes

5. Is the manuscript presented in an intelligible fashion and written in standard English?

Reviewer #3: Yes

6. Review Comments to the Author

Reviewer #3: Thank you for the opportunity to review this thoughtful and timely manuscript. The study introduces R2Play, a novel multi-domain concussion assessment tool that addresses an important gap in return-to-play (RtoP) decision-making among youth. The manuscript is clearly written, well-structured, and meets the general expectations for a proof-of-concept evaluation. The integration of qualitative and quantitative methods is a strength, and the user-centered design process is well-documented.

However, I have concerns regarding the rigor and sufficiency of the statistical analysis, which is why I selected “No” for Question 3. While descriptive summaries are appropriate for a preliminary study, the lack of even basic inferential testing (e.g., paired comparisons, confidence intervals) makes it difficult to assess whether observed trends are meaningful or simply due to chance. For example, changes in multi-task cost scores and usability ratings are discussed as if they are interpretable, yet no thresholds or benchmarks are provided to help the reader understand the relevance or magnitude of these changes.

In addition, some participants had missing data (e.g., heart rate and performance metrics), but there is no clear explanation of how these cases were handled or how they may have influenced the results. The authors also present data from a small, healthy convenience sample, which is fine at this stage, but this limitation should be more directly acknowledged when drawing conclusions about potential clinical value.

Lastly, the mixed-methods design is a strength, but the integration of findings across data types could be more tightly woven. For instance, how did clinician-reported usability or sport-likeness correspond to specific performance data? Even brief insights into how the qualitative and quantitative findings converge would add interpretive depth.

In summary, I believe this is a promising and much-needed contribution to the field. I encourage the authors to further clarify their statistical approach, temper some of the interpretations, and consider how future iterations can build more robust evidence to support R2Play’s clinical application.

7. PLOS authors have the option to publish the peer review history of their article (what does this mean? ). If published, this will include your full peer review and any attached files.

**Do you want your identity to be public for this peer review?** For information about this choice, including consent withdrawal, please see our Privacy Policy .

Reviewer #3: **Yes: ** Victor Ifechukwude Agboli

  **Figure resubmission:**   While revising your submission, we strongly recommend that you use PLOS’s NAAS tool (https://ngplosjournals.pagemajik.ai/artanalysis) to test your figure files. NAAS can convert your figure files to the TIFF file type and meet basic requirements (such as print size, resolution), or provide you with a report on issues that do not meet our requirements and that NAAS cannot fix. 

After uploading your figures to PLOS’s NAAS tool - https://ngplosjournals.pagemajik.ai/artanalysis, NAAS will process the files provided and display the results in the "Uploaded Files" section of the page as the processing is complete. If the uploaded figures meet our requirements (or NAAS is able to fix the files to meet our requirements), the figure will be marked as "fixed" above. If NAAS is unable to fix the files, a red "failed" label will appear above. When NAAS has confirmed that the figure files meet our requirements, please download the file via the download option, and include these NAAS processed figure files when submitting your revised manuscript. **Reproducibility:**  To enhance the reproducibility of your results, we recommend that authors of applicable studies deposit laboratory protocols in protocols.io, where a protocol can be assigned its own identifier (DOI) such that it can be cited independently in the future. Additionally, PLOS ONE offers an option to publish peer-reviewed clinical study protocols. Read more information on sharing protocols at https://plos.org/protocols?utm_medium=editorial-email&utm_source=authorletters&utm_campaign=protocols

---

## [Decision Letter · Decision Letter 2]

29 Sep 2025

Multi-method proof-of-concept evaluation for R2Play: a novel multi-domain return-to-play assessment tool for concussion.

PDIG-D-25-00163R2

Dear Dr. Shore,

We're pleased to inform you that your manuscript has been judged scientifically suitable for publication and will be formally accepted for publication once it meets all outstanding technical requirements.

Within one week, you'll receive an e-mail detailing the required amendments. When these have been addressed, you'll receive a formal acceptance letter and your manuscript will be scheduled for publication.

An invoice for payment will follow shortly after the formal acceptance. To ensure an efficient process, please log into Editorial Manager at https://www.editorialmanager.com/pdig/ click the 'Update My Information' link at the top of the page, and double check that your user information is up-to-date. For billing related questions, please contact billing support at https://plos.my.site.com/s/.

Kind regards,

Shannon Freeman, PhD

Academic Editor

PLOS Digital Health

Additional Editor Comments (optional):

Reviewers' comments:

Reviewer's Responses to Questions

**Comments to the Author**

1. If the authors have adequately addressed your comments raised in a previous round of review and you feel that this manuscript is now acceptable for publication, you may indicate that here to bypass the “Comments to the Author” section, enter your conflict of interest statement in the “Confidential to Editor” section, and submit your "Accept" recommendation.

Reviewer #3: All comments have been addressed

2. Does this manuscript meet PLOS Digital Health’s publication criteria ? Is the manuscript technically sound, and do the data support the conclusions? The manuscript must describe methodologically and ethically rigorous research with conclusions that are appropriately drawn based on the data presented.

Reviewer #3: Yes

3. Has the statistical analysis been performed appropriately and rigorously?

Reviewer #3: Yes

4. Have the authors made all data underlying the findings in their manuscript fully available (please refer to the Data Availability Statement at the start of the manuscript PDF file)?

Reviewer #3: Yes

5. Is the manuscript presented in an intelligible fashion and written in standard English?

PLOS Digital Health does not copyedit accepted manuscripts, so the language in submitted articles must be clear, correct, and unambiguous. Any typographical or grammatical errors should be corrected at revision, so please note any specific errors here.

Reviewer #3: Yes

6. Review Comments to the Author

Please use the space provided to explain your answers to the questions above. You may also include additional comments for the author, including concerns about dual publication, research ethics, or publication ethics. (Please upload your review as an attachment if it exceeds 20,000 characters)

Reviewer #3: The authors have satisfactorily addressed the concerns raised in the previous review, resulting in a clearer and more methodologically sound manuscript. The revisions strengthen the study by improving the clarity of the research objectives, refining the methodological explanations, and enhancing the overall presentation of findings. Ethical considerations are now more explicitly discussed, the statistical analyses are reported with greater transparency, and the discussion section provides a balanced interpretation that acknowledges limitations while emphasizing the study’s contributions. The manuscript is now well-structured, clearly written, and meets the standards for publication. I recommend acceptance.

7. PLOS authors have the option to publish the peer review history of their article (what does this mean? ). If published, this will include your full peer review and any attached files.

**Do you want your identity to be public for this peer review?** For information about this choice, including consent withdrawal, please see our Privacy Policy .

Reviewer #3: Yes: Victor Ifechukwude Agboli
